# New Napyradiomycin Analogues from *Streptomyces* sp. Strain CA-271078

**DOI:** 10.3390/md18010022

**Published:** 2019-12-26

**Authors:** Daniel Carretero-Molina, Francisco Javier Ortiz-López, Jesús Martín, Daniel Oves-Costales, Caridad Díaz, Mercedes de la Cruz, Bastien Cautain, Francisca Vicente, Olga Genilloud, Fernando Reyes

**Affiliations:** 1Fundación MEDINA, Centro de Excelencia en Investigación de Medicamentos Innovadores en Andalucía, Avda. del Conocimiento 34, 18016 Armilla (Granada), Spain; daniel.carretero@medinaandalucia.es (D.C.-M.); jesus.martin@medinaandalucia.es (J.M.); daniel.oves@medinaandalucia.es (D.O.-C.); caridad.diaz@medinaandalucia.es (C.D.); mercedes.delacruz@medinaandalucia.es (M.d.l.C.); bastien.cautain@medinaandalucia.es (B.C.); francisca.vicente@medinaandalucia.es (F.V.); olga.genilloud@medinaandalucia.es (O.G.); 2Doctoral Programme in Pharmacy (B15.56.1), Doctoral School in Health Sciences, University of Granada, 52005 Granada, Spain

**Keywords:** napyradiomycins, marine actinomycetes, structural elucidation, antimicrobial activity, cytotoxicity

## Abstract

As part of our continuing efforts to discover new bioactive compounds from microbial sources, a reinvestigation of extracts of scaled-up cultures of the marine-derived *Streptomyces* sp. strain CA-271078 resulted in the isolation and structural elucidation of four new napyradiomycins (**1**–**3**, **5**). The known napyradiomycin SC (**4**), whose structural details had not been previously described in detail, and another ten related known compounds (**6**–**15**). The structures of the new napyradiomycins were characterized by HRMS and 1D- and 2D-NMR spectroscopies and their relative configurations were established through a combination of molecular modelling with *nOe* and coupling constants NMR analysis. The absolute configuration of each compound is also proposed based on biosynthetic arguments and the comparison of specific rotation data with those of related compounds. Among the new compounds, **1** was determined to be the first non-halogenated member of napyradiomycin A series containing a functionalized prenyl side chain, while **2**–**4** harbor in their structures the characteristic chloro-cyclohexane ring of the napyradiomycin B series. Remarkably, compound **5** displays an unprecedented 14-membered cyclic ether ring between the prenyl side chain and the chromophore, thus representing the first member of a new class of napyradiomycins that we have designated as napyradiomycin D1. Anti-infective and cytotoxic properties for all isolated compounds were evaluated against a set of pathogenic microorganisms and the HepG2 cell line, respectively. Among the new compounds, napyradiomycin D1 exhibited significant growth-inhibitory activity against methicillin-resistant *Staphylococcus aureus*, *Mycobacterium tuberculosis*, and HepG2.

## 1. Introduction

The napyradiomycins are a large class of unique meroterpenoids with different halogenation patterns whose structures consist of a semi-naphthoquinone chromophore, a prenyl unit attached at C-4a that is cyclized to form a tetrahydropyran ring in most cases and a monoterpenoid subunit attached to C-10a [1,2,3,4,5,6]. Many of the structural variants within this family reside in this C-10a attached side chain. Hitherto, about 50 napyradiomycin derivatives (NPDs) have been discovered. They have been sorted into three different types according to their structural features: Type A (NPD-As) with a linear terpenoid side chain; Type B (NPD-Bs) where the side chain is cyclized to form a cyclohexane ring; and Type C (NPD-Cs), whose monoterpenoid subunit is cyclized between C7 and C10a of the naphthoquinone core to form a 14-membered ring [1,2,3,4,5,6,7,8,9,10,11,12,13]. Compounds belonging to this structural class display a wide range of biological activities, including cytotoxic and antibiotic properties, as well as ATPases inhibition or estrogen receptor antagonization [14,15,16,17]. Napyradiomycins were first isolated from the soil-derived bacterium *Chainia rubra* in Japan in 1986 (later transferred to the genus *Streptomyces*) [1,18]. As time passed, a series of congeners were isolated from different actinomycetes including marine-derived strains [3,9].

Multi-drug resistance is one of the emergent threats in the healthcare area due to the loss of effective activity of some drugs against multi-resistant bacteria [19]. The discovery and development of new and safer sources of antibiotics has therefore become an essential matter. Microorganisms from the marine environment are an important source of structurally diverse and biologically active secondary metabolites, as evidenced by the growing number of new marine natural products isolated yearly from different biological sources, but research into the pharmacology of marine organisms is limited and most of it remains unexplored [20,21].

MEDINA’s collection of actinomycetes and filamentous fungi is one of the biggest microbial collections worldwide and contains about ten percent of marine actinomycetes isolated from sediments samples collected along the seafloor [22]. As part of our continuous efforts to discover new bioactive compounds from microbial sources, we initiated a more in-depth study of scaled-up cultures of *Streptomyces* sp. CA-271078, a marine-derived actinomycete strain producing MDN-0170 in whose extracts we had also observed the presence of minor structurally related napyradiomycin congeners by LC/MS analysis [23]. Herein, we report the isolation, structure elucidation, and biological activities of four new napyradiomycin congeners (**1**–**3,5**) together with the description of the spectroscopic features of the known napyradiomycin SC (**4**), all isolated from the ethyl acetate extract of a culture broth of this strain.

## 2. Results

### 2.1. Isolation and Taxonomy of the Producing Microorganism

The isolation and taxonomy of the producing strain, CA-271078, were reported previously based on nearly-complete 16S rRNA gene sequences (1359 bp, 94.2% coverage) [23]. The data obtained strongly indicated that strain CA-271078 is a member of the genus *Streptomyces* and showed the closest relatedness with *Streptomyces aculeolatus* NBRC 14824(T) (99.34% similarity).

### 2.2. Extraction, Dereplication, and Bioassay-Guided Isolation

The producing strain CA-271078 was fermented at 28 °C in 3 L of R358 medium for 6 days. Extraction with an equal volume of acetone and evaporation of the organic solvent after centrifugation and filtration to discard the mycelial debris afforded an acetone crude extract, which was subsequently subjected to liquid–liquid extraction with EtOAc (3 × 1 L). Analysis of this extract by LC/HRMS revealed the presence of some compounds that were not included in our in-house microbial natural products library [24] nor in the Chapman and Hall Dictionary of Natural Products [25].

The extract was subsequently chromatographed on a reversed-phase C18 column using a gradient of acetonitrile in water to afford five fractions: A–E. LC-DAD-HRMS analysis allowed us to establish that all these fractions contained possible known and bioactive NPDs bearing chlorine (according to their isotopic pattern), such as napyradiomycin B6 [26] and 18-hydroxynapyradiomycin A1 [6]. Additionally, these fractions also contained minor amounts of related NPDs whose molecular formulae suggested their novelty as natural products since they were not found in the Dictionary of Natural Products [25]. Further chromatographic separation on semipreparative reversed phase HPLC using a gradient of CH_3_CN/H_2_O, allowed us to isolate fifteen compounds (Appendix A). Napyradiomycins A3 (**1**), B7a (**2**), B7b (**3**), and D1 (**5**) (Figure 1) were identified as new compounds based on ESI-TOF and NMR analyses. Additionally, we have also isolated napyradiomycin SC (**4**) [27] (Figure 1) and its hitherto undescribed spectroscopic features will be reported here. Finally, the spectroscopic data of the rest of the compounds isolated (**6**–**15**) were identical to those previously reported for MDN-0170 (**6**) [23], 3-chloro-6, 8-dihydroxy-8-α-lapachone (**7**) [10], 3-chloro-6-hydroxy-8-methoxy-α-lapachone (**8**) [6], napyradiomycin B6 (**9**) [26], 18-hydroxynapyradiomycin A1 (**10**) [6], napyradiomycin A2a (**11**) [6], napyradiomycin A2b (**12**) [6], napyradiomycin B4 (**13**) [8], napyradiomycin B2 (**14**) [18], and napyradiomycin B5 (**15**) [26] (Appendix A).

### 2.3. Structural Elucidation

Compound **1** was obtained as a white powder and was assigned the molecular formula C_25_H_30_O_7_ (11 degrees of unsaturation) by analysis of the sodium adduct present in its ESI-TOF mass spectrum (*m*/*z* 465.1878 [M + Na]^+^, calcd. for C_25_H_30_NaO_7_^+^, 465.1884) (Appendix A). The UV absorption pattern (Appendix A) with maxima at 258, 314, and 362 nm along with the IR spectrum of **1** (broad absorption bands at around 3294 cm^−1^ for multiple hydroxyl groups, and at 1701 cm^−1^ for a conjugated carbonyl functionality) suggested that this compound possessed the dihydronaphthoquinone moiety typically present in napyradiomycin metabolites.

The ^1^H NMR spectrum of **1** (Table 1 and Appendix A) in DMSO-*d_6_* exhibited three deshielded signals attributable to two aromatic protons at δ_H_ 6.57 ppm (1H, d, 2.1, H-7) and 6.84 ppm (1H, d, 2.1, H-9), and one downflield olefinic proton signal at δ_H_ 6.89 ppm (1H, d, 6.4, H-4). This suggested the presence of a trisubstituted double bond attached to an electron deactivating group. Signals for one olefin methine at δ_H_ 4.98 ppm (1H, br t, 7.25, H-12), three methylene protons at δ_H_ 2.43 ppm (2H, m, H-11), δ_H_ 1.74 ppm, 1.83 ppm (2H, m, H-14) and δ_H_ 1.31 ppm (2H, m, H-15), and two *sp^2^* methylene protons at δ_H_ 4.71 ppm and 4.85 ppm (2H, br s, H-18) were also present. On the other hand, four methyl groups in the aliphatic region at δ_H_ 1.61 ppm (3H, s, H-19), δ_H_ 1.27 ppm (3H, s, H-20), δ_H_ 0.88 ppm (3H, s, H-21), δ_H_ 1.23 ppm (3H, s, H-22), and one exchangeable OH signal (δ_H_ 12.52 ppm), could also be differentiated. The ^13^C NMR spectrum of **1** (Table 2 and Appendix A) exhibited 25 signals: two carbonyl signals at δ_C_ 189.7 and 195.5 ppm, two phenolic carbons (δ_C_ 164.7 and 165.7 ppm), ten *sp^2^* methine or quaternary carbon signals resonating between δ_C_ 107.7 and 148.3 ppm, one *sp^2^* methylene at δ_C_ 110.3 ppm, one *sp^3^* oxygenated quaternary carbon a at δ_C_ 82.2 ppm, two oxygenated methine carbons at δ_C_ 65.8 and 73.4 ppm, as well as other seven aliphatic methylene or methyl carbon signals with chemical shifts below δ_C_ 40.4 ppm. Summing up, according to a heteronuclear single quantum coherence spectroscopy (HSQC) experiment (multiplicity edited) (Appendix A), the ^13^C NMR spectrum evidenced the presence of 4 methyl, 4 methylene, 6 methine, and 11 quaternary carbons. These NMR spectroscopic data suggested that compound **1** was structurally related to the napyradiomycin family of antibiotics, more precisely to dihydronapthoquinones with a 10-carbon monoterpenoid branched side chain (NPDs A series) [4]. A comparison of the NMR spectroscopic data of **1** with those of napyradiomycin A2a/A2b [6] revealed that **1** only differed from napyradiomycin A2a/A2b in the substitution at C-3, with the replacement of the chlorine atom in the latter compound by a hydroxy group in **1** and the presence of an additional olefinic bond at C-4/C-4a. This double bond was easily located since the signals for H_2_-4 were lost and replaced by a *sp^2^* methine signal at δ_H_ 6.89 ppm. Consequently, the ^13^C NMR spectrum (Table 2 and Appendix A) showed two new olefinic carbons for C-4 and C-4a at δ_C_ 134.7 ppm (CH) and δ_C_ 137.6 ppm (C). A comprehensive analysis of 2D NMR data (Figure 2 and Appendix A) allowed the full planar structure of **1** to be assigned, being the first member of the napyradiomycin A-series bearing a hydroxy group rather than a chlorine at position C-3 of the dihydropyran ring (Figure 1).

The relative stereochemistry of the dihydropyran ring of compound **1** (Figure 3) was assigned by interpretation of ROESY data (Appendix A) and the coupling constants observed in its ^1^H NMR spectrum (Table 1, Appendix A) in combination with molecular modelling using Chem3D 12.0 Pro. The biosynthetic route for all napyradiomycins described to date was also considered [3,28,29]. The almost equally intense ROESY correlations observed between H-3 and both geminal methyl groups (C-20 and C-21) together with a coupling constant value of 6.4 Hz between protons H-3 and H-4 (Figure 3), in good agreement with a dihedral angle of 28.7° measured in the energy-minimized molecular model (Appendix A), confirmed the relative configuration at C-3 of the dihydropyran ring in **1**. Assuming an *R* absolute configuration at C-10a based on the common biosynthetic origin described for the napyradiomycin A series described to date, the absolute configuration at C-3 was also proposed to be *R*. Finally, the low amount of **1** obtained prevented the determination of the absolute configuration of the chiral center at C-16 using Mosher´s approaches.

Compounds **2** and **3** were isolated as white powders and their molecular formula was determined to be C_25_H_30_Cl_2_O_6_ on the basis of HRMS measurements (Appendix A) (ESI-TOFMS *m/z* 479.1389 [M + H − H_2_O]^+^ for compound **2** and 479.1394 [M + H − H_2_O]^+^ for compound **3,** calcd. for C_25_H_29_^35^Cl_2_O_5_^+^, 479.1387). This formula requires 10 degrees of unsaturation. Their UV, IR, and NMR spectroscopic data were almost identical and shared common features. The UV absorption pattern (Appendix A) along with the IR spectrum of both (broad absorption bands at around 3294 cm^−1^ for multiple hydroxyl groups, and at 1701 cm^−1^ for a conjugated carbonyl functionality) suggested that these compounds also possessed the dihydronaphthoquinone moiety present in napyradiomycin metabolites. Comprehensive analysis of 1D and 2D NMR data of compounds **2** (Table 1 and Table 2; Appendix A) and **3** (Table 1 and Table 2; Appendix A) revealed a strong similarity between them and evidenced that they were epimers at C-3 position of the dihydropyran ring.

Interpretation of 2D NMR data of compound **2** allowed all the protons and carbons to be assigned and according to a HSQC experiment (multiplicity edited) (Appendix A), we distinguished the presence of 5 methyl, 3 methylene, 6 methine, and 11 quaternary carbons. These NMR spectroscopic data suggested that compound **2** was structurally related to dihydronapthoquinones with the monoterpenoid substituent cyclized to a 6-membered ring (NPDs B series) [4]. The dihydronaphtoquinone ring was constructed based on HMBC correlations (Figure 4 and Appendix A) from H-9 (δ_H_ 6.88 ppm) to C-5a, C-7, C-8, and C-10, and from OH-6 (δ_H_ 12.66 ppm) to C-5a, C-6, and C-7. In addition, correlations in the HMBC experiment between the two geminal methyl groups H_3_-18 and H_3_-19 (δ_H_ 0.94 and δ_H_ 1.40 ppm) and C-2 and C-3, and between the olefinic proton H-4 at δ_H_ 6.73 ppm and C-2, C-3, C-4a, and C-10a indicated the presence of a dihydropyran ring and confirmed the position of a chlorine substituent at C-3 (δ_H_ 4.98, s; δ_C_ 59.9, CH) and a trisubstituted double bond at Δ^4,4a^. Analysis of the overall NMR data set (Table 1 and Table 2; Appendix A) for the monoterpene unit (C-11 to C-17 and C-20, C-21, and C-22) in **2** showed that it was a cyclohexane ring with a chair conformation. Based on these NMR features, the planar structure of **2** was established (Figure 4). A similar analysis of the NMR data set (Table 1 and Table 2; Appendix A) for compound **3** rendered the same planar structure and evidenced the close structural similarity of **2** and **3** with the previously reported antibiotic CNQ525.510A, which is additionally methylated at C-7 [3].

The relative stereochemistry of **2** and **3** was assigned by analysis of NOESY and ROESY NMR data (Appendix A) and the coupling constants observed in their ^1^H NMR spectra. NOESY correlations between the H_3_-21 methyl protons and the methine protons H-12 and H-16 showed these protons were on the same face of the cyclohexane ring in both compounds (Figure 5). A ROESY correlation between the H_3_-20 and H_3_-22 methyl groups determined their axial orientation on the bottom face of the ring. Finally, based on the existence of a strong ROESY correlation, H-12 and H-16 were positioned in a 1,3-diaxial position on the top face of the ring (Figure 5).

Regarding the dihydropyran ring, we used the vicinal ^3^*J*_HH_ spin–spin coupling constants and key NOESY correlations to establish the relative configuration of this moiety. The coupling constant between the olefinic proton at H-3 and its vicinal proton H-4 in **2** had a value of 1.6 Hz (Figure 6), which is in good agreement with a dihedral angle of 91.4° measured in the energy-minimized molecular model (Appendix A). Furthermore, only one NOESY correlation is observed between H-3 and one of the geminal methyl groups (H_3_-19), due to the antiperiplanar position of the H_3_-18 methyl with respect to H-3 (Figure 6 and Appendix A). Thus, the relative configuration on the dihydropyran ring for compound **2** was confirmed as depicted in Figure 1. Conversely, the almost equally intense and strong *nOe* correlations observed between H-3 and both geminal methyl groups (C-18 and C-19) in the spectra of **3** (Figure 6 and Appendix A), together with a coupling constant value of 6.9 Hz between protons H-3 and H-4 in accordance with a dihedral angle of 25.2° measured in the energy-minimized molecular model (Appendix A), confirmed the opposite relative configuration at C-3 for compound **3**.

The absolute configurations of **2** and **3** were assumed to be the same as those reported for all the napyradiomycins in the B series. Apart from the common biosynthetic origin, this proposal was additionally supported by the comparison of specific rotation data. The sign of the specific rotation for compound **2** is negative (${\left[\alpha \right]}_{25}^{D}$ −41.0°, *c* 0.4, MeOH), as is that reported for CNQ525.510A [3], which is a C-7 methylated version of **2**. On the contrary, the sign of ${\left[\alpha \right]}_{25}^{D}$ value for the epimeric compound **3** is positive (${\left[\alpha \right]}_{25}^{D}$ +32.0°, *c* 0.13, MeOH). The previously reported B-type napyradiomycin MDN-0170 [23] displays the same relative configuration at C-3 (but bearing a hydroxy group instead of a chlorine atom), and showed a positive value of specific rotation, providing evidence of the same absolute configuration.

Compound **4** was obtained as a white powder. The complex pattern of the ion clusters in the adducts of its ESI mass spectrum (Appendix A) clearly indicated the presence of two chlorine atoms in the molecule. ESI-TOFMS analysis (Appendix A) suggested the molecular formula C_25_H_32_^35^Cl_2_O_7_ (*m/z* 497.1498 [M + H − H_2_O]^+^, calcd. for C_25_H_31_^35^Cl_2_O_6_^+^, 497.1492), indicating nine degrees of unsaturation. Analysis of combined ^1^H and ^13^C NMR spectroscopic data (Table 1 and Table 2; Appendix A) showed signals similar to those of napyradiomycin metabolites. The presence of a tetrahydropyran ring fused to the dihydronaphthoquinone moiety in **4** was clearly evidenced from the absence of the olefinic proton signal at C-4 present in compounds **1**–**3**, now replaced by two new diastereotopic protons at δ_H_ 2.12 and 2.19 at that position. Furthermore, the presence of a hydroxy substituent at C-4a, easily assigned based on HMBC correlations (Appendix A) of that OH-4a at δ_H_ 6.74 with C-4, C-4a, C10, and C-10a that corroborates the presence of this tetrahydropyran ring. Comprehensive NMR analyses allowed all protons and carbons to be assigned, and compound **4** was confirmed to be napyradiomycin SC, whose structure was incompletely reported by Kamimura and co-workers in a Japanese patent in 1997 [27]. Details on how the structure was assigned, spectroscopic data, and the absolute stereochemistry of compound **4** were never reported. 

The relative stereochemistry of **4** was assigned by analysis of NOESY data (Figure 7 and Appendix A). Correlations of the protons on the cyclohexane ring were identical to those observed in the NOESY/ROESY experiments for **2** and **3**. Key NOESY correlations between the exchangeable OH signal at C-4a with both H-11 protons and the methine H-12 indicated that the tetrahydropyran ring was *cis*-fused to the dihydroquinone, as observed for other napyradiomycins. A correlation observed between the OH-4a hydroxy group and the methine H-3 indicated that these were both oriented on the bottom face of the tetrahydropyran ring. The typical axial-equatorial coupling constants measured between H-3 (*J* = 11.8, 4.4 Hz) and the methylene proton pair H_2_-11 are in good agreement with dihedral angles of –172.6° and –55.3° measured in the energy-minimized molecular model (Appendix A) and confirm that the tetrahydropyran ring was in a chair form, identical to the configuration of this ring in the crystal structure of napyradiomycin B4 [8], a methylated version of compound **4** bearing a chlorine instead of a hydroxy group at C-4a. The full stereostructure of napyradiomycin B4 was assigned by X-ray diffraction methods [8] and hence, the absolute setereochemistry is assumed identical to that of napyradiomycin B4 based on comparable specific rotation values. The sodium D line specific rotations of compound **4** (${\left[\alpha \right]}_{25}^{D}$ −43.0°, *c* 0.4, MeOH) and napyradiomycin B4 (${\left[\alpha \right]}_{25}^{D}$ −190°, *c* 0.031, CHCl_3_) are both negative, providing evidence that both compounds should possess the same absolute configuration. 

Compound **5** was obtained as a white powder and was assigned the molecular formula C_25_H_28_Cl_2_O_5_ by evaluation of HRMS data (Appendix A) (ESI-TOFMS *m/z* 479.1384 [M + H]^+^, calcd. for C_25_H_29_^35^Cl_2_O_5_^+^, 479.1387). The overall analysis of its NMR spectroscopic data (Table 1 and Table 2; Appendix A) clearly showed that the compound possesses the characteristic dihydronaphthoquinone moiety of napyradiomycins, fused to a tetrahydropyran ring and bearing a linear monoterpenoid attached to its C-10a carbon. The ^1^H NMR spectrum of **5** (Table 1, Appendix A) exhibited, among other signals, four singlet methyl groups (δ_H_ 1.11, 1.32, 1.37, and 1.52 ppm), two doublets from aromatic protons (δ_H_ 7.04 and 7.18 ppm), one methine proton geminal to a chlorine atom (δ_H_ 4.40 ppm) and two signals from olefinic methine groups (δ_H_ 4.26 and 4.91 ppm). Interestingly, compound **5** lacks the characteristic deshielded singlet signal of OH-6. HMBC correlations (Appendix A) from H-3 (δ_H_ 4.40 ppm) to C-19/C-20, C-2, and C-4a, and from H_2_-4 (δ_H_ 2.52, 2.25 ppm) to C-2, C-3, C-4a, C-5, and C-10a defined the presence of the archetypal tetrahydropyran ring with two chlorine substituents at C-3 and C-4a positions reported for other napyradiomycins such as 18-hydroxynapyradiomycin A1 (**10**) [6], napyradiomycin A2a (**11**) [6], napyradiomycin A2b (**12**) [6], or napyradiomycin B4 (**13**) [8]. COSY NMR spectroscopic data (Appendix A) allowed the identification of three key proton spin systems within the monoterpenoid moiety: H_2_-11_/_H-12, H_2_-14/-H_2_-15/H-16, and H_2_-18 (Figure 8). These three proton sequences were connected as a linear monoterpenoid side chain by interpretation of HMBC correlations (Figure 8 and Appendix A) from H_3_-21 to C-12, C-13, and C-14, and from H_3_-22 to C-16, C-17, and C-18. As expected, additional HMBC correlations from both H_2_-11 and H-12 clearly established the attachment of this side chain to C-10a of the dihydronaphthoquinone moiety (Figure 8). All these assignments accounted for ten of the eleven degrees of unsaturation and indicated that **5** was composed of a tetracyclic ring system, evidencing that **5** is closely related to napyradiomycin C1 and other compounds within the C series [1,26]. 

The chemical shift of the methylene carbon signal C-18 (δ_C_ 76.3 ppm) strongly supported the presence of an oxygen substituent at this position. Moreover, the deshielding of this carbon signal in comparison with that found for 18-hydroxynapyradiomycin A1 (**10**) (δ_C_ 66.6 ppm) suggested the etherification of the 18-hydroxy groups in **5**. Finally, the strong key HMBC correlation from H_2_-18 to C-6 (δ_C_ 162.7 ppm; Appendix A) allowed us to establish unambiguously the existence of such ether link between C-6 and C-18, which results in a 14-membered bridging macrocycle between C-6 and C-10a (Figure 8). This kind of O-linked cyclization has no precedent within the napyradiomycin metabolites, and therefore compound **5** represents the first member of a new subfamily of napyradiomycins, the D series. Thus, we propose the name napyradiomycin D1 for compound **5**. 

The relative configuration of **5** was determined by inspection of the NOESY correlations and multiplet analysis for some key proton signals (Figure 9). As in napyradiomycin SC (**4**), the values of the coupling constants between H-3 and the pair H_2_-4 (*J* = 11.8, 4.4 Hz) confirmed a chair conformation of the tetrahydropyran ring and the axial orientation of H-3. Although the substitution at C-4a with a chlorine atom in **5** prevents establishing a relative configuration with respect to C-10a, it was assumed to be *cis*, as for all tetrahydropyran-containing napyradiomycins described to date. The geometry of the two double bonds in **5** was assigned as *E* based on the existence of NOESY cross-peaks between H_2_-11/H_3_-21/H_2_-14 and H-15/H_3_-22/H_2_-18, as well as the absence of correlations between H-12/ H_3_-21 and H-16/H_3_-22 (Figure 9a). The absolute configuration of **5** is assumed to be the same as for napyradiomycin C1 considering the comparable specific rotation values and the common biosynthetic origin [1].

Apart from the unambiguous NOESY correlations described above, other sets of cross-peaks were observed for proton signals within the linear monoterpenoid chain (Appendix A). The slight broadening for these signals in the ^1^H NMR spectrum and the presence of such different sets of *nOe* correlations clearly points to a fast (in NMR time-scale) conformational equilibrium in solution for **5**. Not surprisingly, this interconversion might be assisted by the flexibility of the terpenoid chain due to the presence of seven rotatable bonds within it. To better illustrate this, a conformational search using *ConfBuster Web Server*, a recently delivered open-source tool for the conformational analysis of macrocycles was launched [30]. As a result, up to four close-energy conformers were obtained (Appendix A), differing to each other in about 1.5 kcal/mol, which supports the rapid interconversion between them. Interestingly, the minimum-energy conformer proved to be consistent with the most abundant one, as evidenced by the more intense set of NOESY cross-peaks between H_2_-18 (4.67 ppm) and the aromatic proton at H-7 (7.04 ppm), and between the olefinic proton H-16 (4.91 ppm) and H_2_-14 (1.42 ppm) (Figure 9b).

### 2.4. Evaluation of Antimicrobial Activity

#### Antibacterial, Antifungal, and Cytotoxic Activities

Compounds **1**–**15** were evaluated for their antibacterial and antifungal properties against a clinical isolate of methicillin resistant *Staphylococcus aureus* (MRSA), *Mycobacterium tuberculosis*, *Escherichia coli*, *Acinetobacter baumannii*, and *Aspergillus fumigatus* (Table 3). Napyradiomycins **2**, **5**, **7**, **12**, **13**, **14,** and **15** showed antibacterial activities against MRSA with MIC values ranging from 3 to 48 μg/mL. The new napyradiomycin D1 (**5**) was one of the most active compounds and displayed activities comparable to those of napyradiomycin B4 (**13**) and napyradiomycin B5 (**15**). Napyradiomycin A2b (**12**) and napyradiomycin B2 (**14**) exhibited the best antibacterial activities (MIC values of 3–6 μg/mL) among these fifteen napyradiomycins. Except compounds **3**, **6**, **8,** and **10**, the other ten napyradiomycins isolated showed moderate activity against the Gram-positive bacteria *M. tuberculosis* H37Ra with MIC values fluctuating from 12 to 48 μg/mL. None of the compounds exhibited activity against the Gram-negative bacteria *E. coli* ATCC 25922 or *A. baumannii* MB5973 neither against the fungus *A. fumigatus* ATCC 46645.

Eight napyradiomcins, **2**, **5**, **8**, **11**, **12**, **13**, **14**, and **15** showed moderate cytotoxic activities with IC_50_ values below 50 μM against the human liver adenocarcinoma cell line (HepG-2), whereas the other seven had reduced cytotoxicities, with IC_50_ values above this concentration.

## 3. Discussion

The napyradiomycins constitute a large class of unique halogenated meroterpenoids, with around fifty members reported to date, produced by marine and terrestrial *Streptomyces* species [1,2,3,4,5,6,7,8,9,10,11,12,13]. The biosynthetic gene cluster (BGC) of napyradiomycins was first described from *Streptomyces aculeolatus* NRRL 18422 and from the marine derived *Streptomyces*
*sp.* CNQ-525, and its analysis established the link between the presence of three vanadium-dependent haloperoxidases (VHPOs), (NapH1, NapH3, and NapH4) and a chloronium-induced meroterpene cyclization pathway [28]. The full biosynthetic route to these metabolites from three precursors (1,3,6,8-tetrahydroxynaphthalene, dimethylallyl pyrophosphate, and geranyl pyrophosphate), has been recently described and highlights the key role of those VHPO enzymes [29].

The final structures resulting from this biosynthetic pathway, the napyradiomycins, are hybrid terpenoid/polyketide metabolites composed of a semi-naphthoquinone chromophore, a prenyl unit attached to C-4a which is cyclized to produce a tetrahydropyran or dihydropyran ring and a monoterpenoid subunit attached at C-10a, which in turn can be either linear (type A napyradiomycins) or cyclized to 6-membered (type B) or to 14-membered (type C) rings. The different halogenation patterns add complexity and contribute to structural variations of these interesting metabolites [1,2,3,4,5,6,7,8,9,10,11,12,13]. A further proof of the still surprising structural possibilities for these natural products is the herein reported isolation of four new napyradiomycins, A3 (**1**), B7a (**2**), B7b (**3**), and D1 (**5**), showing unusual substitution patterns, inverted configuration of some chiral centers, or unprecedented cyclic bridging links.

Compound **1** is to the best of our knowledge the first example of a napyradiomycin in the A series bearing a hydroxy group instead a chlorine atom at position C-3 of the dihydropyran ring. This substitution pattern has been previously reported for napyradiomycins of types B (MDN-0170) [23] and C [1,26], further displaying the same relative configuration as **1** at this chiral center. The presence of the C-3 hydroxy group with this absolute configuration can be explained considering the precursor of **1** might be the corresponding chlorinated compound at the same position, and that a non-enzymatic S_N_2 nucleophilic substitution with water on the C-3 chloride would result in the production of **1**.

Compounds **2** and **3** are epimers at C-3 of their dihydropyran ring. The relative configuration at this chlorinated position for compound **3** is reported herein for the first time in the napyradiomycin series, since all the natural products of this family found in the literature or databases have the opposite configuration at the same chiral center (i.e., that found for compound **2**) when it has a chlorine substituent. A first tentative explanation for this variant arises from the proposed mechanism of oxidative halogenation and subsequent halonium-induced cyclization in meroterpenoids [28]. Although the vanadium-dependent chloroperoxidase (VCPO) NapH1 has been shown to act in a stereoselective fashion when introducing chlorine atoms in napyradiomycin biosynthetic intermediates, it was also found that the same enzyme catalyzed a non-stereoselective bromination of the same substrate [31]. This result was explained because of the production of a diffusible hypobromous acid, which would depart the active site of the enzyme and then would brominate the substrate in a nonspecific manner. Indeed, the involvement of a hypohalous acid (HOX species) in this mechanism is widely accepted for vanadium-dependent haloperoxidases (VHPOs) from algae and fungi, which do not exhibit specificity [32], while for VCPOs from Streptomycetes it has been postulated that an enzyme-bound chlorine species would make possible the stereoselective halogenation [33,34]. 

The isolation herein of both diastereomeric versions of the same chlorinated product (compounds **2** and **3**) could suggest the participation of a hypochlorous acid mediated chlorination along with the enzyme-assisted mechanism. However, the fact that this nonspecific chlorination has not been observed for any other napyradiomycin derivatives isolated in this work, i.e. from the same culture and therefore under the same conditions, did not support this overall hypothesis.

The other possibility is that the chlorine atom at C-3 in compound **3** may come from a S_N_2 displacement, therefore with inversion of configuration. The only reasonable way for that to occur is that a chloride ion is displacing the C-3 chlorine in compound **2**. Although it is known that chloride is a poor nucleophile (and a fair leaving group), the activation of this allylic position within the dihydropyran ring could provide enough driving force for the reaction to proceed. Despite this epimerization has not been observed for any other dihydropyran ring-containing napyradiomycin (isolated in this work or not), we consider that the nucleophilic substitution to produce compound **3** may be the most plausible hypothesis. 

Finally, compound **5** presents in its structure a 16-membered macrocycle because of a bridging ether link between C-6 of the dihydronaphtoquinone unit and C-18 of the monoterpenoid chain. This kind of ether link is unprecedented within the napyradionmycin family, and therefore compound **5** represents the first member reported of a new series of napyradiomycins that we have designated as napyradiomycin D1. Remarkably, although a similar 15-membered cyclic ether ring has been previously reported for merochlorin C—a compound belonging to another family of meroterpenoid metabolites [35]—the O-linked bridging observed in **5** still represents, to the best of our knowledge, one of the largest ether cyclization events observed for a natural product. Regarding its biosynthesis, since NapH1 has been proved responsible for the halogenation and the formation of a 6-membered cyclic ether ring in napyradiomycins [31], it is tempting to think that the same enzyme could also catalyze this macrocyclization. However, the attachment in napyradiomycin D1 of the ether linkage to a former methyl group without the installation of any chiral chlorinated center vicinal to the linking position is inconsistent with a chloronium-induced cyclization catalyzed by NapH1. The whole genome sequencing of the producer strain CA-271078 and the inspection of its napyradiomycin BGC may further provide valuable data about this and other biosynthetic questions raised in this work.

Although the specific mechanism of action for this family of meroterpenoids is not totally clear [5,14,15], previous studies about the structure activity relationship (SAR) have shown that structural variations among the napyradiomycin metabolites scaffold known so far, such as the different halogenation patterns or the presence or absence of the methyl group at C-7 among others, can attenuate or enhance their biological activities [4,14]. 

Some of the compounds isolated displayed antibacterial activity against MRSA. Notably, the activity of compound **5**, the first in the napiradiomycin D series, was comparable to that of other known compounds isolated in this work. In line with previous results, none of the compounds isolated were found to be active against Gram-negative bacteria or fungi. Additionally, our cytotoxicity data illustrate that, for the napyradiomycin B series, the C-3 chlorinated derivatives (compound **2** and **3**) exhibit a higher cytotoxic activity compared to the hydroxylated analogous at that stereocenter (compound **6**). This cytotoxic is in turn significantly influenced by the absolute stereochemistry of the chlorine group at that position (compound **2** vs. **3**). On the other hand, there is a clear correlation between rising levels of cytotoxicity when the substitution pattern at C-4a varies between hydroxy group (compound **4**), hydrogen (compound **9),** or chlorine (compound **13**).

## 4. Materials and Methods 

### 4.1. General Experimental Procedures

Optical rotations were measured on a Jasco P-2000 polarimeter (JASCO Corporation, Tokyo, Japan). IR spectra were recorded with a JASCO FT/IR-4100 spectrometer (JASCO Corporation) equipped with a PIKE MIRacle^TM^ single reflection ATR accessory. 1D- and 2D-NMR spectra were recorded on a Bruker Avance III spectrometer (500 and 125 MHz for ^1^H and ^13^C NMR, respectively) equipped with a 1.7 mm TCI MicroCryoProbe^TM^ (Bruker Biospin, Fällanden, Switzerland). Chemical shifts were reported in ppm using the signals of the residual solvents as internal reference (δ_H_ 2.51 and δ_C_ 39.5 ppm for DMSO-*d_6_*). LC-UV-MS analysis was performed on an Agilent 1100 (Agilent Technologies, Santa Clara, CA, USA) single quadrupole LC-MS system as previously described [24]. ESI-TOF spectra were acquired using a Bruker maXis QTOF (Bruker Daltonik GmbH, Bremen, Germany) mass spectrometer coupled to an Agilent 1200 LC (Agilent Technologies, Waldbronn, Germany). Medium pressure liquid chromatography (MPLC) was performed on semiautomatic flash chromatography (CombiFlash Teledyne ISCO Rf400x) with a precast reversed-phase column. Semi-preparative HPLC separation was performed on Gilson GX-281 322H2 (Gilson Technologies, USA) with a semi-preparative reversed-phase column (Zorbax SB-C18, 250 × 9.4 mm, 5 μm). Preparative HPLC separation was performed on Gilson GX-281 322H2 (Gilson Technologies, USA) with a reversed-phase column (Zorbax SB-C18, 250 × 21.2 mm, 7 μm). Acetone used for extraction was analytical grade. Solvents employed for isolation were HPLC grade. Molecular models were generated using Chem3D Pro 12.0 (CambridgeSoft, PerkinElmer Informatics, Waltham, MA, USA). The structures were energy-minimized by molecular mechanics with the MM2 force field using as gradient convergence criteria an RMS value of 0.001. Molecular modelling figures were generated with PyMol (W. L. DeLano, The PyMOL Molecular Graphics System, DeLano Scientific LLC, Palo Alto, CA, USA, 2002). Conformational search was performed for 5 with ConfBuster Web Server, using as input the corresponding sdf files generated with Chem3D Pro 12.0 (CambridgeSoft, PerkinElmer Informatics, Waltham, MA, USA).

### 4.2. Taxonomic Identification of the Producing Microorganism

The taxonomic identification of the strain was described in a previous work [23]. 

### 4.3. Fermentation of the Producing Microorganism

A 3 L fermentation of the strain CA-271078 of *Streptomyces* sp. was generated using the conditions described in reference [23].

### 4.4. Extraction and Bioassay Guided Isolation

The fermentation broth (3 L) was extracted with acetone (3 L) under continuous shaking at 220 rpm for 1 h. The mycelium was separated and discarded by centrifugation at 9000 rpm and filtration and the supernatant (ca. 6 L) was concentrated to 3L under reduced pressure of nitrogen stream. The aqueous crude extract was extracted with EtOAc (3 × 0.9 L) to afford a crude extract of 0.178 g. This EtOAc extract was loaded onto a Reversed-Phase C18 (ODS) column (32 × 100 mm) that was eluted with a gradient of acetonitrile in water (35% to 100% ACN in 50 min + 100% ACN in 10 min, 15 mL/min, 18 mL/fraction) to afford 50 fractions. They were combined into five fractions according to their LC-UV-MS profiles and evaporated to dryness in a centrifugal evaporator: fractions A (6.6 mg), B (5.3 mg), C (9.0 mg), D (12.7 mg) and E (17.8 mg). Fractions containing the compounds of interest from this chromatography were further purified by semipreparative reversed-phase HPLC.

Fraction A (6.6 mg) was chromatographed by semipreparative reversed-phase HPLC (Zorbax SB-C18, 9.4 × 250 mm, 5 μm; 3.6 mL/min, UV detection at 210 and 280 nm) with an isocratic elution of 33% CH_3_CN/ 67% H_2_O with 0.1% trifluoroacetic acid over 34 min yielding **1** (0.7 mg, *t*_R_ 18 min) and **6** (2.1 mg. *t*_R_ 25 min).

Fraction B (5.3 mg) was chromatographed by semipreparative reversed-phase HPLC (Zorbax SB-C18, 9.4 × 250 mm, 5 μm; 3.6 mL/min, UV detection at 210 and 280 nm) with an isocratic elution of 35% CH_3_CN/ 65% H_2_O with 0.1% trifluoroacetic acid over 34 min yielding **7** (1.1 mg, *t*_R_ 21.5 min).

Fraction C (9.0 mg) was subjected to reversed-phase semipreparative HPLC (Zorbax SB-C18, 9.4 × 250 mm, 5 μm; 3.6 mL/min, UV detection at 210 and 280 nm) with and isocratic elution of CH_3_CN/ H_2_O 55/45 with 0.1% trifluoroacetic acid over 34 min, yielding **3** (0.7 mg, *t*_R_ 15 min), **4** (0.5 mg, *t*_R_ 22.5 min), **8** (0.5 mg, *t*_R_ 20.5 min) and **9** (0.5 mg, *t*_R_ 24 min).

Fraction D (12.7 mg) was chromatographed by semipreparative reversed-phase HPLC (Zorbax SB-C18, 9.4 × 250 mm, 5 μm; 3.6 mL/min, UV detection at 210 and 280 nm) with a linear gradient of CH_3_CN/ H_2_O with 0.1% trifluoroacetic acid, from 55 to 65 % CH_3_CN over 34 min, yielding **2** (1.3 mg, *t*_R_ 23.5 min), **10** (0.8 mg, *t*_R_ 25.2 min), **11** (0.9 mg, *t*_R_ 27 min), **12** (1.3 mg, *t*_R_ 28.5 min) and **13** (0.9 mg, *t*_R_ 31 min).

Fraction E (12.7 mg) was subjected to reversed-phase semipreparative HPLC (Zorbax SB-C18, 9.4 × 250 mm, 5 μm; 3.6 mL/min, UV detection at 210 and 280 nm) with a linear gradient of CH_3_CN/ H_2_O with 0.1% trifluoroacetic acid, from 65 to 70 % CH_3_CN over 34 min, yielding **5** (1.0 mg, *t*_R_ 21 min), **14** (1.0 mg, *t*_R_ 27.5 min) and **1****5** (1.0 mg, *t*_R_ 32.5 min).

### 4.5. Characterization Data

*Napyradiomycin A3* (**1**): ${\left[\alpha \right]}_{25}^{D}$ +4.0 (*c* 0.38, MeOH); IR (ATR) cm^−1^: 3379, 2977, 1702, 1673, 1619, 1267, 1186, 1136, 1024, 837, 720; (+)-ESI-TOFMS *m/z* 465.1877 [M + Na]^+^ (calcd. for C_25_H_30_NaO_7_^+^, 465.1884), 460.2330 [M + NH_4_]^+^ (calcd. for C_25_H_34_NO_7_^+^, 460.2330), 443,2060 [M + H]^+^ (calcd. for C_25_H_31_O_7_^+^, 443.2064), 425.1958 [M + H − H_2_O]^+^ (calcd. for C_25_H_29_O_6_^+^, 425.1960); ^1^H and ^13^C NMR data see Table 1 and Table 2.

*Napyradiomycin B7a* (**2**): ${\left[\alpha \right]}_{25}^{D}$ −41.0 (*c* 0.4, MeOH); IR (ATR) cm^−1^: 3374, 2930, 1680, 1615, 1377, 1260, 1202, 1137, 1025, 798, 722; (+)-ESI-TOFMS *m/z* 1010.3184 [2M + NH_4_]^+^ (calcd. for C_50_H_64_^35^Cl_4_NO_12_^+^, 1010.3177),514.1753 [M + NH_4_]^+^ (calcd. for C_25_H_34_^35^Cl_2_NO_6_^+^, 514.1758), 479.1389 [M + H − H_2_O]^+^ (calcd. for C_25_H_29_^35^Cl_2_O_5_^+^, 479.1387), 461.1284 [M + H − 2H_2_O]^+^ (calcd. for C_25_H_27_^35^Cl_2_O_4_^+^, 461.1281); ^1^H and ^13^C NMR data see Table 1 and Table 2.

*Napyradiomycin B7b* (**3**): ${\left[\alpha \right]}_{25}^{D}$ +32.0 (*c* 0.13, MeOH); IR (ATR) cm^−1^: 3368, 2980, 1680, 1618, 1455, 1378, 1268, 1204, 1137, 803; (+)-ESI-TOFMS *m/z* 1010.3191 [2M + NH_4_]^+^ (calcd. for C_50_H_64_^35^Cl_4_NO_12_^+^, 1010.3177), 514.1752 [M + NH_4_]^+^ (calcd. for C_25_H_34_^35^Cl_2_NO_6_^+^, 514.1758), 479.1394 [M + H − H_2_O]^+^ (calcd. for C_25_H_29_^35^Cl_2_O_5_^+^, 479.1387), 461.1294 [M + H − 2H_2_O]^+^ (calcd. for C_25_H_27_^35^Cl_2_O_4_^+^, 461.1281), 443.1629 [M + H − 3H_2_O]^+^ (calcd. for C_25_H_25_^35^Cl_2_O_3_^+^, 443.1175); ^1^H and ^13^C NMR data see Table 1 and Table 2.

*Napyradiomycin SC* (**4**): ${\left[\alpha \right]}_{25}^{D}$ −43.0 (*c* 0.4, MeOH); IR (ATR) cm^−1^: 3340, 2976, 1703, 1632, 1615, 1464, 1372, 1264, 1224, 1183, 1082, 1024; (+)-ESI-TOFMS *m/z* 497.1499 [M + H − H_2_O]^+^ (calcd. for C_25_H_31_^35^Cl_2_O_6_^+^, 497.1492), 479.1389 [M + H − 2H_2_O]^+^ (calcd. for C_25_H_28_^35^Cl_2_O_5_^+^, 479.1387); ^1^H and ^13^C NMR data see Table 1 and Table 2. 

*Napyradiomycin D1* (**5**): ${\left[\alpha \right]}_{25}^{D}$ +14.6 (*c* 0.13, MeOH); IR (ATR) cm^−1^: 3358, 2978, 2948, 1671, 1608, 1458, 1371, 1298, 1185, 1136, 1079, 1028, 802; (+)-ESI-TOFMS *m/z* 479.1386 [M + H]^+^ (calcd. for C_25_H_29_^35^Cl_2_O_5_^+^, 479.1387); ^1^H and ^13^C NMR data see Table 1 and Table 2.

### 4.6. Antibacterial and Antifungal Assays Cytotoxic Activities

Compounds **1**–**15** were tested in antimicrobial assays against the growth of Gram-negative (*E. coli* ATCC 25922 and *A*. *baumannii* MB5973) and Gram-positive bacteria (*M. tuberculosis* H37Ra and methicillin-resistant *S. aureus* (MRSA) MB5393), and fungi (*A. fumigatus* ATCC46645) and in a cytotoxicity assay against the human liver adenocarcinoma cell line (HepG2) following previously described methodologies [36,37,38]. Briefly, each compound was serially diluted in DMSO with a dilution factor of 2 to provide 10 concentrations starting at 96 μg/mL for all the antimicrobial assays except for compounds **3** and **8** which started at 64 μg/mL. For the adenocarcinoma cell line each compound was serially diluted in DMSO with a dilution factor of 2 to provide 10 concentrations starting at 150 μg/mL (compounds **2**, **4**, **6**, **7**, and **9**–**15**), 100 μg/mL (compounds **3** and **8**) or 30.0 μg/mL (compounds **1** and **5**). The MIC was defined as the lowest concentration of compound that inhibited ≥ 95% of the growth of a microorganism after overnight incubation. The Genedata Screener software (Genedata, Inc., Basel, Switzerland) was used to process and analyze the data and also to calculate the RZ’ factor, which predicts the robustness of an assay [39]. In all experiments performed in this work the RZ’ factor obtained was between 0.87 and 0.98. 

## 5. Conclusions

In this work, we report the isolation and structural characterization of four new napyradiomycins (**1**–**3**, **5**) and the known napyradiomycin SC (**4**), whose structural details had not been previously described. Additionally, another ten known napyradiomycins or related compounds (**6**–**15**) were also isolated from the same culture broth of the marine-derived *Streptomyces* sp. CA-271078 from MEDINA’s microbial collection. The antibacterial, antifungal and cytotoxic properties of all the compounds isolated were tested. Napyradiomycins B2 (**14**), B4 (**13**), and B5 (**15**) and the new napyradiomycin D1 (**5**) were the most active compounds, exhibiting similar antibacterial activities against MRSA and *M. tuberculosis* H37Ra, as well as comparable cytotoxic activities against the HepG2 tumoral cell line. On the contrary, none of the compound tested showed significant activity against *E. coli*, *A. baumannii*, or *A. fumigatus.*

The four new compounds isolated displayed remarkable structural features. Thus, compound **1** is the first member of napyradiomycins in the A series bearing a hydroxy group rather than a chlorine atom at C-3. The isolation of the new B-type napyradiomycin **3** represents to our knowledge the first example reported with a different relative configuration at the C-3 chlorinated position in napyradiomycins and raises biosynthetic and mechanistic questions to be considered. Compound **5** harbors in its structure an unprecedented 16-membered macrocyclic ether ring between the naphthoquinone moiety and the monoterpenoid chain, thus inaugurating a new class of napyradiomycins, the D series.

The results reported here highlight the wide range of structural possibilities for napyradiomycin metabolites. The further whole genome sequencing of the producer strain CA-271078 and the analysis of the corresponding napyradiomycins gene cluster could bring new data about the biosynthesis of these fascinating natural products.

## Figures and Tables

**Figure 1 marinedrugs-18-00022-f001:**
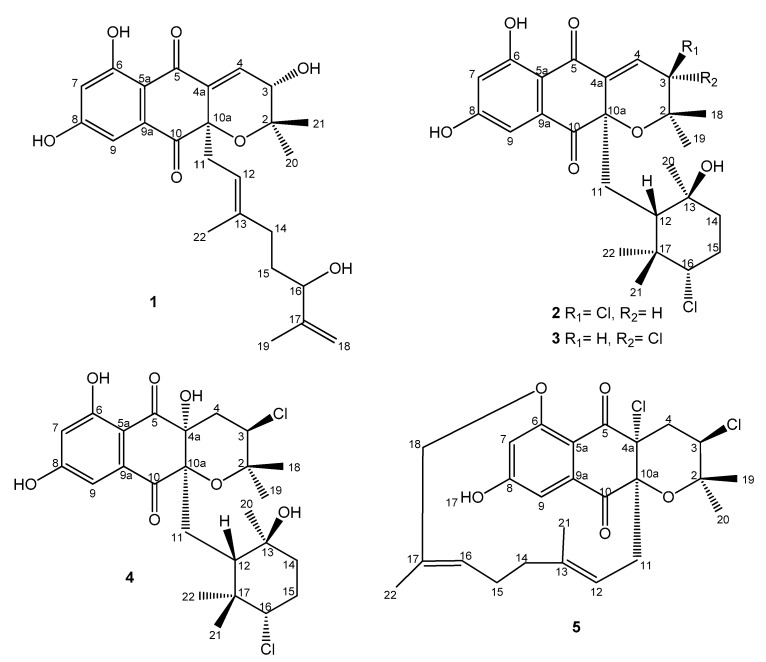
Compounds **1**–**5** isolated from culture broths of *Streptomyces sp*. CA-271078.

**Figure 2 marinedrugs-18-00022-f002:**
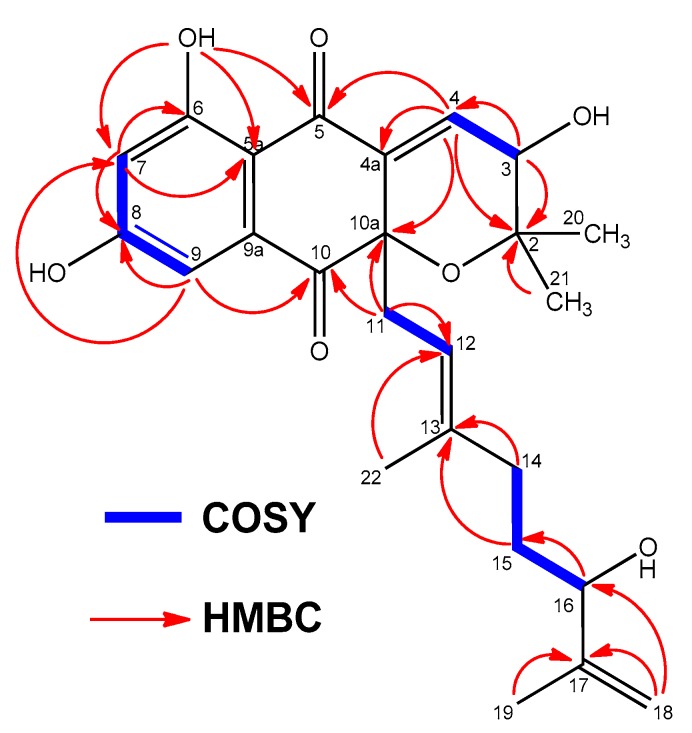
Key COSY and HMBC correlations observed in the spectra of compound **1**.

**Figure 3 marinedrugs-18-00022-f003:**
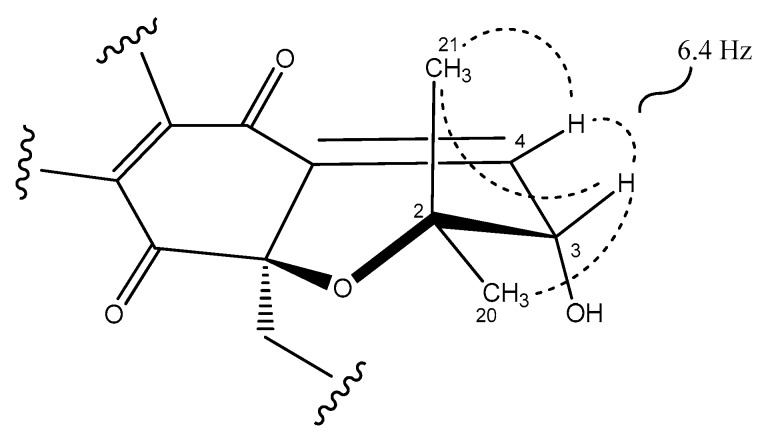
Key ROESY correlations (dashed lines) and coupling constant that determine the relative configuration of the dihydropyran ring in napyradiomycin A3 (**1**).

**Figure 4 marinedrugs-18-00022-f004:**
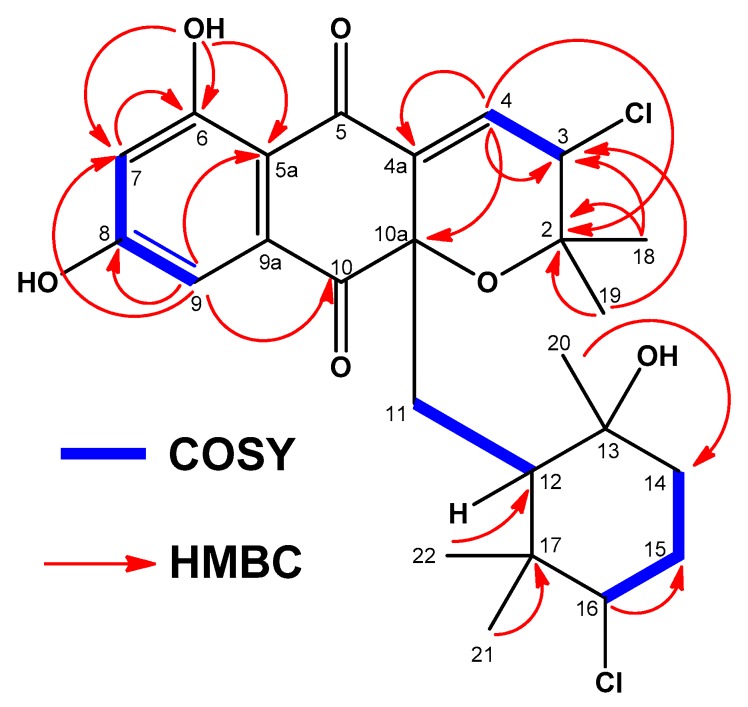
Key COSY and HMBC correlations observed in the spectra of compound **2** or **3**.

**Figure 5 marinedrugs-18-00022-f005:**
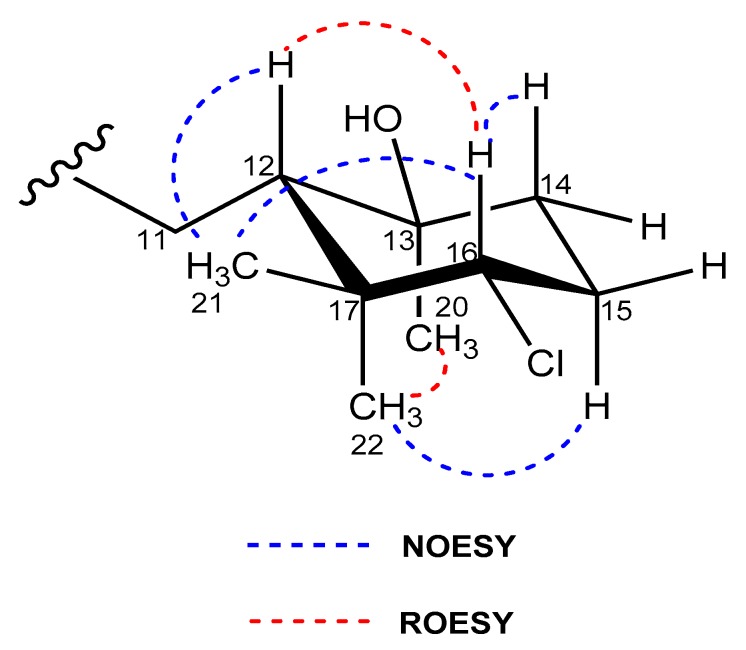
Key NOESY/ROESY correlations that determine the relative configuration of the cyclohexane ring in compounds **2** and **3**.

**Figure 6 marinedrugs-18-00022-f006:**
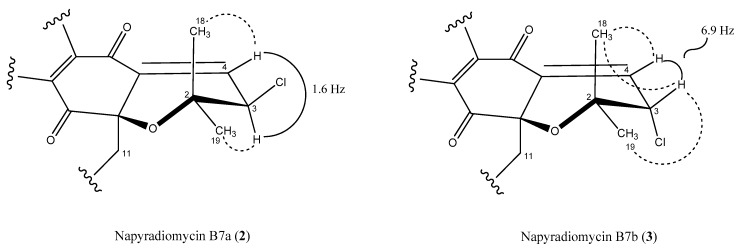
Key NOESY correlations (dashed lines) and coupling constants that determine the relative configuration of the dihydropyran ring in compounds **2** and **3**.

**Figure 7 marinedrugs-18-00022-f007:**
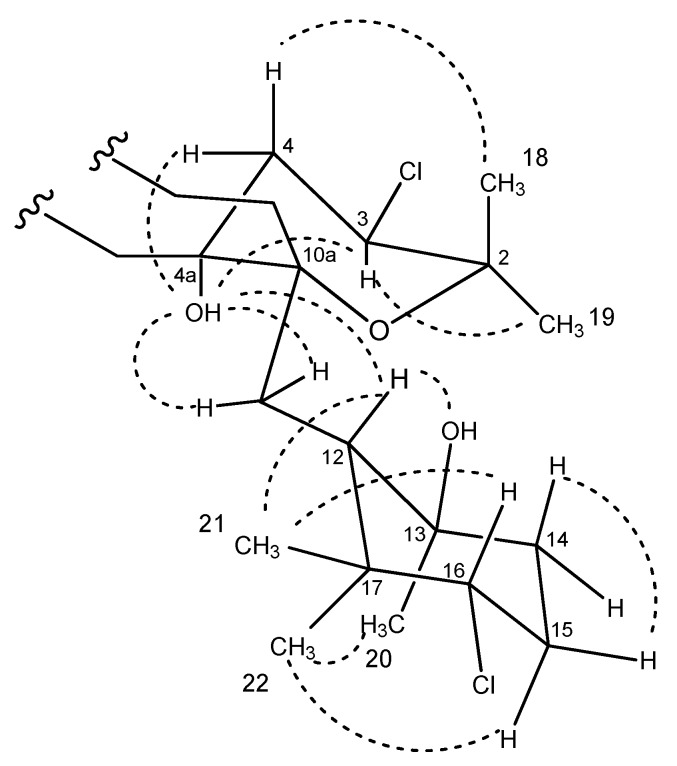
Conformation in solution of compound **4** based upon NOESY analysis.

**Figure 8 marinedrugs-18-00022-f008:**
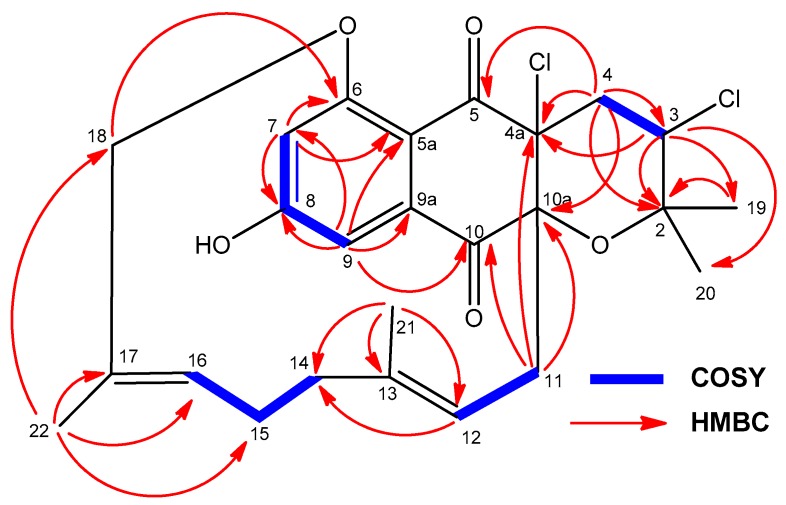
Key COSY and HMBC correlations observed in the spectra of compound **5**.

**Figure 9 marinedrugs-18-00022-f009:**
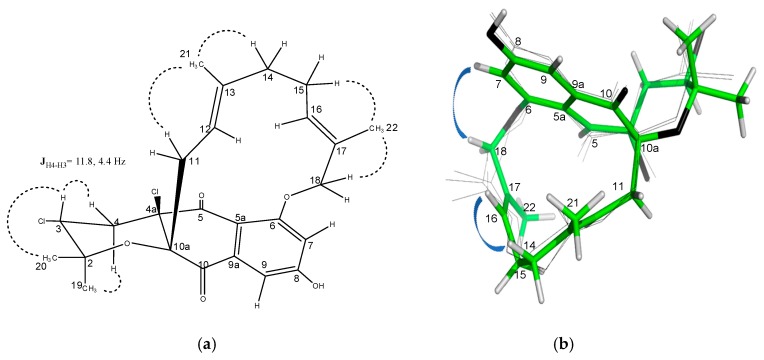
(**a**) Key NOESY correlations (dashed lines) supporting the relative configuration of napyradiomycin D1 (**5**). (**b**) Minimum-energy conformer for **5** and consistent *nOe* cross-peaks.

**Table 1 marinedrugs-18-00022-t001:** ^1^H NMR (500 MHz in DMSO-*d_6_*) data for compounds **1**-**5**.

No.	δ ^1^H (mult, J, Hz)				
	1	2	3	4	5
**2**					
**3**	3.72, d (6.4)	4.98, d (1.6)	4.70, d (6.9)	4.38, dd (11.8, 4.4)	4.40, dd (11.8, 4.4)
**4**	6.89, d (6.4)	6.73, d (1.6)	7.01, d (6.9)	2.12, dd (13.4, 4.4)	2.25 ^a^, dd (14.4, 4.4)
				2.19, dd, (13.4, 11.8)	2.52, dd (14.4, 11.8)
**4a**				6.74, s	
**5**					
**5a**					
**6**	12.52, s	12.66, s	12.56, s	11.94, s	
**7**	6.57, d (2.1)	6.59, d (2.2)	6.60, d (1.9)	6.66, d (2.0)	7.04, d (2.4)
**8**	--	--	--	--	--
**9**	6.84, d (2.1)	6.88, d (2.2)	6.88, d (1.9)	6.99, d (2.0)	7.18, d (2.4)
**9a**					
**10**					
**10a**					
**11**	2.43, m	1.52 ^a^, dd (13.3, 3.4)	1.68, dd (15.2, 2.4)	1.27, br d (16.4, 2.3)	2.26 ^a^, m
		1.85 ^b^, dd (13.3, 3.20)	1.93, dd (15.2, 6.0)	2.32, dd (16.4, 6.0)	2.77, dd (13.5,8.9)
**12**	4.98, br t (7.25)	1.31, br t (3.3)	1.60 ^a^, dd (6.0, 2.4)	1.55, dd (6.0, 2.3)	4.26, br t (8.9)
**13**		4.75, s	4.40, s	5.04, s	
**14**	1.74, m	1.46, m	1.37, m	1.47, m	1.42, m
	1.83, m	1.51 ^a^, m	1.60 ^a^, m	1.67, m	1.93 ^b^, m
**15**	1.31, m	1.70, dd (13.5, 12.1)	1.70, dd (13.2, 12.1)	1.72, m	1.95 ^b^, m
		1.81 ^b^, dd (13.5, 3.8)	1.83, dd (13.2, 3.1)	1.82, m	
**16**	3.78, t (6.2, 6.2)	4.02, dd (12.1, 3.8)	3.77, dd (12.1, 3.1)	3.81, dd (11.9, 3.9)	4.91, br t (11.5, 6.5)
**17**					
**18**	4.71, br s	0.94 ^c^, s	1.02, s	1.15, s	4.67, s
	4.85, br s				4.73, s
**19**	1.61, s	1.40, s	1.42, s	1.35, s	1.37, s
**20**	1.27, s	0.79, s	0.91, s	1.02, s	1.11, s
**21**	0.88, s	0.94 ^c^, s	0.77, s	0.38, s	1.32, s
**22**	1.23, s	0.64, s	0.63, s	0.59, s	1.52, s

^a,b,c^ overlapping signals.

**Table 2 marinedrugs-18-00022-t002:** ^13^C NMR (125 MHz in DMSO-*d_6_*) data for compounds **1**–**5**.

No.	δ ^13^C				
	1	2	3	4	5
**2**	75.4, C	76.1, C	74.8, C	79.1, C	78.2, C
**3**	65.8, CH	59.9, CH	57.3, CH	58.9, CH	59.8, CH
**4**	134.7, CH	136,2 CH	132.1, CH	40.4, CH_2_	41.6 CH_2_
**4a**	137.6, C	136.3, C	137.8, C	79.2, C	81.7, C
**5**	189.7, C	187.7, C	188.3, C	194.4, C	183.4, C
**5a**	110.8, C	110.5, C	110.4, C	107.9, C	116.7, C
**6**	164.7, C	164.8, C	164.8, C	163.9, C	162.7, C
**7**	107.8, CH	108.1, CH	108.1, CH	108.5, CH	113.5, CH
**8**	165.7, C	165.3, C	165.6, C	165.7, C	163.8, C
**9**	107.7, CH	107.7, CH	107.9, CH	107.8, CH	108.3, CH
**9a**	136.5, C	135.9, C	135.9, C	135.1, C	136.4, C
**10**	195.5, C	195.3, C	194.9, C	199.5, C	196.0, C
**10a**	82.2, C	82.9, C	82.6, C	83.7, C	82.9, C
**11**	40.4, CH_2_	37.0, CH_2_	37.8, CH_2_	33.7, CH_2_	41.5, CH_2_
**12**	117.7, CH	50.5, CH	50.1, CH	48.6, CH	116.9, CH
**13**	138.9, C	70.5, C	70.1, C	70.1, C	140.6, C
**14**	35.5, CH_2_	41.2, CH_2_	41.3, CH_2_	41.2, CH_2_	39.7, CH_2_
**15**	32.9, CH_2_	31.1, CH_2_	30.8, CH_2_	30.5, CH_2_	23.7, CH_2_
**16**	73.4, CH	72.2, CH	72.4, CH	71.8, CH	126.5, CH
**17**	148.3, C	40.4, C	40.4, C	40.2, C	129.2, C
**18**	110.3, CH_2_	20.5, CH_3_	25.8, CH_3_	21.6, CH_3_	76.3, CH_2_
**19**	17.8, CH_3_	26.5, CH_3_	26.5, CH_3_	28.7, CH_3_	28.9, CH_3_
**20**	24.9, CH_3_	23.6, CH_3_	24.2, CH_3_	24.4, CH_3_	22.5, CH_3_
**21**	25.7, CH_3_	30.1, CH_3_	29.0, CH_3_	28.4, CH_3_	15.2, CH_3_
**22**	16.1, CH_3_	16.6, CH_3_	16.7, CH_3_	16.0, CH_3_	15.1, CH_3_

**Table 3 marinedrugs-18-00022-t003:** Antibacterial, antifungal, and cytotoxic activities of compounds **1**–**15**.

	MIC (μg/mL)					IC_50_ (μM)
	MRSA	*Mt*	*Ec*	*Ab*	*Af*	HepG-2
1	>96	NT ^a^	>96	>96	NT ^a^	>67.8
2	48	12–24	>96	>96	>96	41.7
3	>64	>64	>64	>64	>64	109.5
4	>96	24–48	>96	>96	>96	263.5
5	12–24	24–48	>96	>96	NT ^a^	14.9
6	>96	>96	>96	>96	>96	277.2
7	48–96	12–24	>96	>96	>96	186.9
8	>64	>64	>64	>64	>64	30.2
9	48–96	12–24	>96	>96	>96	71.2
10	48–96	>96	>96	>96	>96	64.4
11	12–24	48–96	>96	>96	>96	30.4
12	12–24	48–96	>96	>96	>96	28.6
13	12–24	12–24	>96	>96	>96	15.6
14	3–6	24–48	>96	>96	>96	27.1
15	12–24	24–48	>96	>96	>96	40.1

Note: MRSA, Methicillin-resistant *Staphylococcus aureus* MB5393; Mt, *Mycobacterium tuberculosis* H37Ra; Ec, *Escherichia coli* ATCC 25922; Ab, *Acinetobacter baumannii* MB5973; Af, *Aspergillus fumigatus* ATCC 46645. ^a^ NT = not tested.

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
