# Peer review of "New Napyradiomycin Analogues from Streptomyces sp. Strain CA-271078"

_marinedrugs, 2019, doi:10.3390/md18010022_

Round 1

Reviewer 1 Report

The submission from Carretero-Molina et al reports the discovery of a variety of novel napyradiomycin molecules from a previously explored marine Streptomyces sp. strain CA-271078. I appreciated the authors’ discovery and spectroscopic and bioactivity characterization of these novel napyradiomycin molecules. Undeniably, the most interesting molecule is the macrocyclic ether napyradiomycin D1, which is novel for this molecular family. I thought the authors presented strong evidence to support its chemical structure using both spectroscopic and molecular modeling techniques. My major criticisms are largely to do with some of the biosynthetic proposals suggested in the discussion section, in addition to some grammatical and spelling errors. I also have a few minor adjustments regarding how the compounds are numbered, and the macrocyclic ether ring size (I count a 14 membered ether ring).

Despite these critiques, I thoroughly enjoyed the manuscript and recommend that it be accepted after addressing the minor revisions listed below.

Content adjustments:

Line 28 – The use of ‘functionalized prenyl side chain’ isn’t descriptive enough to differentiate compound 1 from other A-type napyradiomycins. Maybe ‘first in the A series with an allylic alcohol in the tricyclic ring?’ Or ‘the first non-halogenated napyradiomycin A-like molecule?’ I would include something more descriptive here.

Line 30 – I count a 14-membered macrocyclic ether ring (10a, 11, 12, 13, 14, 15, 16, 17, 18, O6, 6, 5a, 9a, 10). This is still an impressive and novel metabolite within the napyradiomycin family, and worthy of the new designation of napyradiomycin D1. However I don’t see a 16-membered ether ring, and I believe IUPAC nomenclature is consistent with a 14-membered ring (should select the path to make the smallest possible ring system).

Line 47 – Instead of ‘branched side chain’ (which implies a branched terpenoid like in the merochlorin series of meroterpenoids) I would use a ‘linear’ terpenoid sidechain.

Line 75 – I was confused about what ‘nearly-complete 16S’ meant? Does this refer to the fact that the 16S rRNA sequence was nearly the same as the Streptomyces aculeolatus NBRC 14824(T) strain? Or that only a portion of the 16S rRNA was sequenced? Some clarification here would be appreciated.

Line 101/Figure 1 – In general the compounds are all numbered consistently except for the methyl groups. Sometimes the methyl/methylene groups at the end of the former geranyl moiety are labeled 18/19 (in compound 1), 21/22 (in compounds 2-4) or 18/22 (in compound 5). The same ambiguity exists with the methyls in the pyran ring (sometimes 20/21, 18/19, 19/20). Right now lines 18-22 in Tables 1 and 2 are not a good source of comparison across this molecular series because they correlate to different carbons in in the molecule. For the sake of consistency and easy comparison, I’d suggest using one numbering system throughout all of the molecules so that Tables 1 and 2 are more useful.

Figure 1 – There is a number 14 under oxygen 1 in structures 1-3, and the tetrahydropyran ‘O’ in compound 4 is a different font/size than the other heteroatoms

Line 149 – I think the references are off in this paragraph, as references 27 and 28 don’t speak to napyradiomycin biosynthesis. I think the intended references are 29 and 30? I also think reference 27 should be included in section 2.2 when discussing the known but uncharacterized napyradiomycin SC.

Table 1 – Compound 5 has a δH for H8 of 12.17? Is this accurate or a typo? It doesn’t seem to be present in the 1H NMR spectra (Figure S36) and is usually indicative of the H-bonded phenolic H6. This chemical shift is key for identifying this novel metabolite so I would ensure that it is correct.

Line 157 – At the author’s discretion, they could mention that both hydroxyl epimers have been seen at this position (napyradiomycins A2a and A2b from Motohashi et al, J. Nat. Prod. 2008 – currently reference 6), and so it is possible that both exist in their molecule as well.

Line 159/Figure 3 – The C3-OH group is reported here to have a ROESY correlation with C19-methyl but I can’t see a reported chemical shift for this hydroxyl in Table 1.

Line 213/Figure 6 – Compounds 2 and 3 are called MDN-0242 and MDN-0241 respectively underneath their structures instead of the nomenclature napyradiomycin B7a and B7b as listed throughout the remainder of the text. I would keep the nomenclature consistent.

Line 305/Figure 9a/b – Is it possible to orient these molecules the same way or number key atoms? Either of those would better emphasize your main correlations in the two structures.

Lines 322-336 – There is inconsistent usage of compound names and numbers in this section. I would still use the numbering system even if the compound name is mentioned (eg, The new napyradiomycin D1 (5) was one of the most . . . ) just to help make a stronger connection with the chemical structures. This is especially helpful for the molecules that were rediscovered in this study (napyradiomycin B4, B5, etc) that weren’t discussed as thoroughly as the novel 5 metabolites.

Lines 332-333 – Napyradiomycins have typically shown good antimicrobial activity against Gram-positive bacteria but not against Gram-negative bacteria or fungi. I would make that more clear in the text, and maybe adjust the columns of Table 3 to put the most napyradiomycin-sensitive organisms first (MRSA/Mt) to further emphasize this point.

Lines 334-336 – Is 27 μM negligible? It’s about half as active as the napyradiomycins with ‘moderate cytotoxic activities’. I would be careful drawing the line at 20 μM and discounting the other compounds with comparable (albeit reduced) cytotoxicities compared to the most potent molecules. There seems to be a more clear barrier at maybe 50 or 100 μM, but the way this is currently presented seems to equally compare compound 14 with a 27 μM IC50 with one that has ten-fold reduced potency (277 μM).

Line 348 – The authors should either use a lowercase ‘n’ in NapH1/NapH3/NapH4 if they want to refer to the VHPO genes, or keep it as written but not italicize them to refer to the VHPO proteins.

Lines 368-369 – An enzymatic hydroxylation is possible, but I think a much more likely scenario is a non-enzymatic SN2 displacement with water on the C-3 chloride. Any of the C-3 epimers (whether it be a hydroxyl or chloride group) only occur after the installation of the adjacent double bond between C-4 and C-4a, which would activate this C-3 allylic chloride for nucleophilic substitution. To further support this point, no epimers to the best of my knowledge have been isolated with the C-4 methylene intact. This suggests to me that this epimerization is non-enzymatic and a result of the inherent chemical reactivity of the dihydropyran version of these molecules.

Lines 394-398 – I disagree with this statement; these vanadium-haloperoxidase enzymes have shown impressive regio- and enantiospecificity when interrogated with in vitro substrates. If diffusible HOCl was being produced, there would be less specific aryl ring or geranyl moiety halogenation. I think the allylic nucleophilic substitution argument described in the previous point, this time using a chloride anion as the nucleophile, makes more sense for the production of this particular molecule.

Line 407 – NapH1 should not be italicized, and I disagree with this particular biosynthetic hypothesis. While VHPO catalysis (NapH1) does form an ether linkage and the pyran ring, it does so by having an oxygen atom attack an enzyme-catalyzed chloronium ion at the terminal double bond of the prenyl moiety. This creates a chiral halogenated center vicinal to the ether linkage. A similar mechanism is proposed for VHPO Mcl40 in merochlorin biosynthesis to produce the macrocyclic compound merochlorin C. This mechanism is inconsistent for the cyclic ether system observed in napyradiomycin D1, as the ether linkage is attached to a former methyl group without the installation of any vicinal halides. So I would be careful implicating NapH1 in this particular macrocyclization reaction.

Lines 412-421 – There is a fairly large difference between the biological activity of epimers 2 and 3 in both antibacterial and cytotoxicity assays, something that isn’t heavily discussed in any context. I would mention the importance of the stereochemistry at this position for biological activity.

Lines 509-510 – I would specify which strains are Gram-positive and Gram-negative.

Line 531 – H37Ra shouldn’t be italicized

Spelling mistakes/grammatical errors:

I think there was a conversion error when submitting the paper as nearly all Greek symbols seem to be missing in my PDF copy (noticeable from section 2.3 on).

Line 22 – phrasing: and another ten related known. . .

Line 32/33 – phrasing: . . . properties for all isolated compounds were evaluated. . .

Line 32 (and throughout) – compound names (napyradiomycins) should not be capitalized.

Line 35 – spelling: methicillin

Line 44/45 – rephrase: Many of the structural variants within this family reside in this C-10a attached side chain.

Line 54 – spelling: marine-derived

Line 87 – spelling: NPDs (instead of NDPs as written)

Line 140 – phrasing: for compounds 1-5

Line 142 – phrasing: for compounds 1-5

Line 201 – spelling: dihydropyran

Line 211 – space between: 25.2° measured

Lines 230-235 – phrasing: very long and confusing sentence

Lines 273-274 – phrasing: allowed the identification of three key proton spin systems within the monoterpenoid moiety: H2-11/ etc. . . .

Line 278 – spelling: dihydronaphthoquinone

Line 288 – spelling: has no precedent within the . .

Line 316 – spelling: lower case ‘k’ in kcal

Lines 323-324 – phrasing: . . . antibacterial and antifungal properties against clinical isolates of Escherichia coli, . . .

Line 356 – spelling: napyradiomycins

Line 449 – spelling: taxonomic

Line 509 – spelling: Gram-negative

Line 526 – phrasing: Additionally, another ten known. . . .

Line 542 – phrasing: “non-yet spent” doesn’t make sense to me, replace that with something else.

Author Response

We deeply appreciate the comments and suggestions made by this reviewer, which have greatly improved some parts of our manuscript. To address his/her concerns, we have implemented the following changes:

Line 28 – The use of ‘functionalized prenyl side chain’ isn’t descriptive enough to differentiate compound 1 from other A-type napyradiomycins. Maybe ‘first in the A series with an allylic alcohol in the tricyclic ring?’ Or ‘the first non-halogenated napyradiomycin A-like molecule?’ I would include something more descriptive here. Addressed.

Line 30 – I count a 14-membered macrocyclic ether ring (10a, 11, 12, 13, 14, 15, 16, 17, 18, O6, 6, 5a, 9a, 10). This is still an impressive and novel metabolite within the napyradiomycin family, and worthy of the new designation of napyradiomycin D1. However I don’t see a 16-membered ether ring, and I believe IUPAC nomenclature is consistent with a 14-membered ring (should select the path to make the smallest possible ring system). Addressed.

The reviewer is right. Considering the path to make the smallest possible ring, the new molecule has a 14-membered ether ring. This correction has been applied wherever was necessary in the manuscript.

Consistently, the text in lines 404-405 of the original document:

“…the O-linked bridging observed in 5 still represents, to the best of our knowledge, the largest olefin-involved ether cyclization observed for a natural product

has been replaced by the following in the revised version:

“…the O-linked bridging observed in 5 still represents, to the best of our knowledge, one of the largest ether cyclization events observed for a natural product.

Line 47 – Instead of ‘branched side chain’ (which implies a branched terpenoid like in the merochlorin series of meroterpenoids) I would use a ‘linear’ terpenoid sidechain. Addressed.

Line 75 – I was confused about what ‘nearly-complete 16S’ meant? Does this refer to the fact that the 16S rRNA sequence was nearly the same as the Streptomyces aculeolatus NBRC 14824(T) strain? Or that only a portion of the 16S rRNA was sequenced? Some clarification here would be appreciated. Addressed.

Nearly-complete 16S indicates that the length of the 16S rDNA gene amplified and sequenced (1359 base pairs) is close to the full reported 16S rDNA from Streptomyces aculeolatus (1442 base pairs). Therefore, we have a 94.24% coverage ((1359/1442)*100) and a 99.34% similarity within that coverage. Line 75 has been modified to make it more clear.

Line 101/Figure 1 – In general the compounds are all numbered consistently except for the methyl groups. Sometimes the methyl/methylene groups at the end of the former geranyl moiety are labeled 18/19 (in compound 1), 21/22 (in compounds 2-4) or 18/22 (in compound 5). The same ambiguity exists with the methyls in the pyran ring (sometimes 20/21, 18/19, 19/20). Right now lines 18-22 in Tables 1 and 2 are not a good source of comparison across this molecular series because they correlate to different carbons in in the molecule. For the sake of consistency and easy comparison, I’d suggest using one numbering system throughout all of the molecules so that Tables 1 and 2 are more useful.

We appreciate this comment from the reviewer, but considering that the structures of the new compounds have also been explained in their corresponding sections, renumbering all the molecules in a different way would suppose a significant modification in several parts of the text, with a high possibility of introducing mistakes and would complicate excessively the revision of the manuscript. We therefore have only implemented some minor changes to at least consistently number compounds 2-4, which share a common skeleton.

Figure 1 – There is a number 14 under oxygen 1 in structures 1-3, and the tetrahydropyran ‘O’ in compound 4 is a different font/size than the other heteroatoms Addressed.

Line 149 – I think the references are off in this paragraph, as references 27 and 28 don’t speak to napyradiomycin biosynthesis. I think the intended references are 29 and 30? I also think reference 27 should be included in section 2.2 when discussing the known but uncharacterized napyradiomycin SC.  Addressed.

The reviewer is right

Table 1 – Compound 5 has a δH for H8 of 12.17? Is this accurate or a typo? It doesn’t seem to be present in the 1H NMR spectra (Figure S36) and is usually indicative of the H-bonded phenolic H6. This chemical shift is key for identifying this novel metabolite so I would ensure that it is correct. Addressed.

It was a typo, that signal is not present. It has been removed from Table 1.

Line 157 – At the author’s discretion, they could mention that both hydroxyl epimers have been seen at this position (napyradiomycins A2a and A2b from Motohashi et al, J. Nat. Prod. 2008 – currently reference 6), and so it is possible that both exist in their molecule as well.

Certainly, the epimers napyradiomycins A2a and A2b were indistinguishable using NMR data, but they were easily separated by ODS HPLC. We understand that we do not have a mixture of both C-16 diastereomers for 1 since applying similar HPLC methods to prove it we have just obtained one peak.

Line 159/Figure 3 – The C3-OH group is reported here to have a ROESY correlation with C19-methyl but I can’t see a reported chemical shift for this hydroxyl in Table 1. Addressed.

It was a typo, we can not observe that ROESY correlation between C3-OH group and C-19 methyl in Figure 3. It has been removed from Figure 3. The ROESY correlation we wanted to indicate was between H-3 and C-20 methyl group. (This is now represented in Figure 3). Please also note that there was a mistake in the numbering of the methyl groups in figure 3 that has been corrected.
There is not a reported chemical shift for C3-OH in Table 1 because we cannot observe a signal for that hydroxy group.

Line 213/Figure 6 – Compounds 2 and 3 are called MDN-0242 and MDN-0241 respectively underneath their structures instead of the nomenclature napyradiomycin B7a and B7b as listed throughout the remainder of the text. I would keep the nomenclature consistent. Addressed.

Line 305/Figure 9a/b – Is it possible to orient these molecules the same way or number key atoms? Either of those would better emphasize your main correlations in the two structures. Addressed.

We have numbered key atoms in Figure 9b.

Lines 322-336 – There is inconsistent usage of compound names and numbers in this section. I would still use the numbering system even if the compound name is mentioned (eg, The new napyradiomycin D1 (5) was one of the most . . . ) just to help make a stronger connection with the chemical structures. This is especially helpful for the molecules that were rediscovered in this study (napyradiomycin B4, B5, etc) that weren’t discussed as thoroughly as the novel 5 metabolites. Addressed.

Lines 332-333 – Napyradiomycins have typically shown good antimicrobial activity against Gram-positive bacteria but not against Gram-negative bacteria or fungi. I would make that more clear in the text, and maybe adjust the columns of Table 3 to put the most napyradiomycin-sensitive organisms first (MRSA/Mt) to further emphasize this point. Addressed.

Lines 332-33 have been modified and the columns of Table 3 have been adjusted to make it more clear. Additional comments on bioactivity of the compounds have been added at the end of the discussion section

Lines 334-336 – Is 27 μM negligible? It’s about half as active as the napyradiomycins with ‘moderate cytotoxic activities’. I would be careful drawing the line at 20 μM and discounting the other compounds with comparable (albeit reduced) cytotoxicities compared to the most potent molecules. There seems to be a more clear barrier at maybe 50 or 100 μM, but the way this is currently presented seems to equally compare compound 14 with a 27 μM IC50 with one that has ten-fold reduced potency (277 μM). Addressed.

We have made a clear distinction between cytotoxicities above and below 50 μM

Line 348 – The authors should either use a lowercase ‘n’ in NapH1/NapH3/NapH4 if they want to refer to the VHPO genes, or keep it as written but not italicize them to refer to the VHPO proteins. Addressed.

Lines 368-369 – An enzymatic hydroxylation is possible, but I think a much more likely scenario is a non-enzymatic SN2 displacement with water on the C-3 chloride. Any of the C-3 epimers (whether it be a hydroxyl or chloride group) only occur after the installation of the adjacent double bond between C-4 and C-4a, which would activate this C-3 allylic chloride for nucleophilic substitution. To further support this point, no epimers to the best of my knowledge have been isolated with the C-4 methylene intact. This suggests to me that this epimerization is non-enzymatic and a result of the inherent chemical reactivity of the dihydropyran version of these molecules.

We agree that the hydroxy group at C-3 in compound 1 most likely comes from nucleophilic displacement with water of the chlorine atom on the corresponding chlorinated precursor. We also agree that the allylic C-3 position generated after the installation of the double bond between C-4 and C-4a (dihydropyran ring) would be activated for this nucleophilic attack. However, if we would consider the existence of both C-3 epimeric versions of the putative chlorinated dihydropyran precursor (and this co-existence is proved in this work for compounds 2 and 3), we would have to admit that this displacement should occur with either inversion (SN2) or retention (SN1) of configuration, depending on the stereochemistry considered for those chlorinated precursors (see lines 377-380 in the original manuscript). Anyway, since the typically observed absolute configuration for C-3 chlorinated position is (R), and that of the same center bearing a hydroxy group in 1 is (S), we accept that this hydroxylation most likely occurs through SN2 displacement with water of the chloride atom in the corresponding chlorinated precursor.

We have modified the manuscript, so where it was written:

“As a first interpretation, and leaving aside stereochemical considerations, the presence of this hydroxy group could be explained considering that the precursor of 1 might be the corresponding chlorinated compound at the same position, and that an enzymatic hydroxylation via nucleophilic substitution would result in the production of 1.

Now it is written:

“The presence of the C-3 hydroxy group with this absolute configuration can be explained considering that the precursor of 1 might be the corresponding chlorinated compound at the same position, and that a non-enzymatic SN2 nucleophilic substitution with water on the C-3 chloride would result in the production of 1.”

Lines 394-398 – I disagree with this statement; these vanadium-haloperoxidase enzymes have shown impressive regio- and enantiospecificity when interrogated with in vitro substrates. If diffusible HOCl was being produced, there would be less specific aryl ring or geranyl moiety halogenation. I think the allylic nucleophilic substitution argument described in the previous point, this time using a chloride anion as the nucleophile, makes more sense for the production of this particular molecule.

Indeed, VHPOs from Streptomycetes, and more precisely NapH1, have shown high stereoselectivity in the chlorination under in vitro conditions, as we mention in the original manuscript (see lines 383-385; [31]). However, it was reported in the same article that this enzyme catalyzed a nonspecific bromination and it was hypothesized that a diffusible HOBr species could be the responsible for the lack of specificity in the bromination [31]. Since during this work we isolated both diastereomeric versions of the same chlorinated product (compounds 2 and 3), we postulated that a hypochlorous acid-mediated chlorination could be co-occurring (yielding both compounds 2 and 3 in a nonselective manner) along with the cited enzyme-bound chlorine species halogenation (yielding only compound 2 in a stereoselective fashion). Anyway, we are now aware that the fact that this (or any other) nonspecific chlorination has not been observed for any other napyradiomycin isolated in this work (that is, from the same culture and therefore under the same conditions), does not support this overall hypothesis.

The other possibility, suggested by the reviewer, is that the chlorine atom at C-3 in compound 3 may come from a SN2 displacement, therefore with inversion of configuration. The only feasible way for that to occur is that a chloride ion is displacing the C-3 chlorine in compound 2. However, it is known that chlorine is a poor nucleophile as well as a fair leaving group, and therefore the reaction might not have enough driving force to proceed, even on this activated allylic position. Moreover, this epimerization has not been observed for any other dihydropyran ring-containing napyradiomycin (isolated in this work or not), so it does not seem to be a so inherent chemical feature of this kind of systems, as suggested by the reviewer.

Anyway, all arguments considered together, we finally think that the hypothesis of the nucleophilic substitution for the production of compound 3 may be more plausible than the involvement of a diffusible HOCl species. Consistently, this part of the discussion has been modified in the following way:

The paragraph between lines 373-380 was eliminated:

“.. As commented above,…. / / …. depending on the stereochemistry considered for those chlorinated precursors

The paragraph between lines 380-397 was modified. The original text:

 “Another perspective to explain these variations arises from the proposed mechanism of oxidative halogenation and subsequent halonium-induced cyclization in meroterpenoids [29]. Although the vanadium-dependent chloroperoxidase (VCPO) NapH1 has been shown to act in a stereoselective fashion when introducing chlorine atoms in napyradiomycin biosynthetic intermediates, it was also found that the same enzyme catalyzed a non-stereoselective bromination of the same substrate, and two stereoisomeric bromohydrines were found as presumable precursors of this latter transformation [31]. This result was explained as a consequence of the production of a diffusible hypobromous acid, which would depart the active site of the enzyme and then would brominate the substrate in a nonspecific manner. Indeed, the involvement of a hypohalous acid (HOX species) in this mechanism is widely accepted for vanadium-dependent haloperoxidases (VHPOs) from algae and fungi, which do not exhibit specificity [32], while for VCPOs from Streptomycetes it has been postulated that an enzyme-bound chlorine species would make possible the stereoselective halogenation [33].

The isolation herein of both diastereomeric versions of the same chlorinated product (compounds 2 and 3) suggest that a hypochlorous acid -mediated oxidative chlorination should also be considered. Moreover, the participation of this HOCl species could further promote the different substitution patterns (hydroxy, chlorine) at C-3 position for the different napyradiomycin metabolites reported in this work.

has been replaced with this:

“A first tentative explanation for this variant arises from the proposed mechanism of oxidative halogenation and subsequent halonium-induced cyclization in meroterpenoids [29]. Although the vanadium-dependent chloroperoxidase (VCPO) NapH1 has been shown to act in a stereoselective fashion when introducing chlorine atoms in napyradiomycin biosynthetic intermediates, it was also found that the same enzyme catalyzed a non-stereoselective bromination of the same substrate [31], and this result was explained as a consequence of the production of a diffusible hypobromous acid, which would depart the active site of the enzyme and then would brominate the substrate in a nonspecific manner. Indeed, the involvement of a hypohalous acid (HOX species) in this mechanism is widely accepted for vanadium-dependent haloperoxidases (VHPOs) from algae and fungi, which do not exhibit specificity [32], while for VCPOs from Streptomycetes it has been postulated that an enzyme-bound chlorine species would make possible the stereoselective halogenation [33].

The isolation herein of both diastereomeric versions of the same chlorinated product (compounds 2 and 3) could suggest the participation of a hypochlorous acid -mediated chlorination along with the enzyme-assisted mechanism. However, the fact that this nonspecific chlorination has not been observed for any other napyradiomycin derivatives isolated in this work, i. e. from the same culture and therefore under the same conditions, does not support this overall hypothesis.”

And the following paragraph has been added:

“The other possibility is that the chlorine atom at C-3 in compound 3 may come from a SN2 displacement, therefore with inversion of configuration. The only reasonable way for that to occur is that a chloride ion is displacing the C-3 chlorine in compound 2. Although it is known that chloride is a poor nucleophile (and a fair leaving group), the activation of this allylic position within the dihydropyran ring could provide enough driving force for the reaction to proceed. Despite this epimerization has not been observed for any other dihydropyran ring-containing napyradiomycin (isolated in this work or not), we consider that the nucleophilic substitution to produce compound 3 may be the most plausible hypothesis”

Line 407 – NapH1 should not be italicized, and I disagree with this particular biosynthetic hypothesis. While VHPO catalysis (NapH1) does form an ether linkage and the pyran ring, it does so by having an oxygen atom attack an enzyme-catalyzed chloronium ion at the terminal double bond of the prenyl moiety. This creates a chiral halogenated center vicinal to the ether linkage. A similar mechanism is proposed for VHPO Mcl40 in merochlorin biosynthesis to produce the macrocyclic compound merochlorin C. This mechanism is inconsistent for the cyclic ether system observed in napyradiomycin D1, as the ether linkage is attached to a former methyl group without the installation of any vicinal halides. So I would be careful implicating NapH1 in this particular macrocyclization reaction.

We appreciate the comment from the reviewer and have modified the manuscript accordingly, so where it was written:

“…, it is reasonable to think that the same enzyme could also catalyze this macrocyclization.”

now is written:

“…, it is tempting to think that the same enzyme could also catalyze this macrocyclization. However, the attachment in napyradiomycin D1 of the ether linkage to a former methyl group without the installation of any chiral chlorinated center vicinal to the linking position is inconsistent with a chloronium-induced cyclization catalyzed by NapH1.”

Lines 412-421 – There is a fairly large difference between the biological activity of epimers 2 and 3 in both antibacterial and cytotoxicity assays, something that isn’t heavily discussed in any context. I would mention the importance of the stereochemistry at this position for biological activity. Addressed.

We appreciate the comment from the reviewer and we have added a paragraph at the endo of the discussion section highlighting this fact .

Lines 509-510 – I would specify which strains are Gram-positive and Gram-negative. Addressed.

Line 531 – H37Ra shouldn’t be italicized. Addressed.

Spelling mistakes/grammatical errors:

I think there was a conversion error when submitting the paper as nearly all Greek symbols seem to be missing in my PDF copy (noticeable from section 2.3 on). Addressed.

Line 22 – phrasing: and another ten related known. . . Addressed.

Line 32/33 – phrasing: . . . properties for all isolated compounds were evaluated. . . Addressed.

Line 32 (and throughout) – compound names (napyradiomycins) should not be capitalized. Addressed.

Line 35 – spelling: methicillin. Addressed.

Line 44/45 – rephrase: Many of the structural variants within this family reside in this C-10a attached side chain. Addressed.

Line 54 – spelling: marine-derived. Addressed.

Line 87 – spelling: NPDs (instead of NDPs as written). Addressed.

Line 140 – phrasing: for compounds 1-5. Addressed.

Line 142 – phrasing: for compounds 1-5. Addressed.

Line 201 – spelling: dihydropyran. Addressed.

Line 211 – space between: 25.2° measured. Addressed.

Lines 230-235 – phrasing: very long and confusing sentence. To be addressed.

We appreciate the comment from the reviewer and have modified the manuscript accordingly, so where it was written:

“The absence in 4 of the olefinic proton signal at C-4 present in compounds 1-3, replaced by two new diastereotopic protons at dH 2.12 and 2.19 at that position along with the presence of a hydroxy substituent at C-4a, easily assigned based on HMBC correlations (Fig. S32) of that OH-4a at dH 6.74 with C-4, C-4a, C10 and C-10a, clearly evidenced that the compound possesses a tetrahydropyran ring fused to the dihydronaphthoquinone moiety.”

now it reads:

“The presence of a tetrahydropyran ring fused to the dihydronaphthoquinone moiety in 4 was clearly evidenced from the absence of the olefinic proton signal at C-4 present in compounds 1-3, now replaced by two new diastereotopic protons at dH 2.12 and 2.19 at that position. Furthermore, the presence of a hydroxy substituent at C-4a, easily assigned based on HMBC correlations (Fig. S32) of that OH-4a at dH 6.74 with C-4, C-4a, C10 and C-10a also corroborates the presence of this tetrahydropyran ring”

Lines 273-274 – phrasing: allowed the identification of three key proton spin systems within the monoterpenoid moiety: H2-11/ etc. . . . Addressed.

Line 278 – spelling: dihydronaphthoquinone. Addressed.

Line 288 – spelling: has no precedent within the . .. Addressed.

Line 316 – spelling: lower case ‘k’ in kcal. Addressed.

Lines 323-324 – phrasing: . . . antibacterial and antifungal properties against clinical isolates of Escherichia coli, . . . Addressed.

Line 356 – spelling: napyradiomycins Addressed.

Line 449 – spelling: taxonomic Addressed.

Line 509 – spelling: Gram-negative Addressed.

Line 526 – phrasing: Additionally, another ten known. . . . Addressed.

Line 542 – phrasing: “non-yet spent” doesn’t make sense to me, replace that with something else. Addressed.

Reviewer 2 Report

General Comments

The manuscript deals with the isolation, structure elucidation and biological activities of four new members of the napyradiomycin family of antibiotics and still unkown structural details of the known napyradiomycin SC, and in addition other ten related compounds produced by the marine Streptomyces sp. strain CA-271078. The paper is of scientific importance, it is very well written, the results are clearly described, and it can be accepted for publication after minor revision.

Specific Comments

Lines 32, 88, 131, 132, 138, 294, 302, 328: Napyradiomycins should be written homogeneously in lower case letters.

Line 52/53: Change “Genus Streptomyces” to “genus Streptomyces”.

Lines 338-340: The organisms should be written in italics.

Line 509: “Gram negative” should be changed to “Gram-negative”.

Lines 640, 645,651: The title of the article in references 26, 28 and 30 must be changed to lower case letters, as done in all other references.

Author Response

We acknwledge the effort and time invested by this reviewer to check our manuscript. All his comments regarding minor changes have been implemented in the revised version uploaded.